# Autologous cell transplantation for treatment of colorectal aganglionosis in mice

Weikang Pan[1,2,5], Ahmed A. Rahman[1,5], Takahiro Ohkura ®[1], Rhian Stavely[1], Kensuke Ohishi[1,3], Christopher Y. Han[1], Abigail Leavitt[1], Aki Kashiwagi[1], Alan J. Burns[1,4], Allan M. Goldstein[1] & Ryo Hotta ®[1] ✉

Neurointestinal diseases cause significant morbidity and effective treatments are lacking. This study aims to test the feasibility of transplanting autologous enteric neural stem cells (ENSCs) to rescue the enteric nervous system (ENS) in a model of colonic aganglionosis. ENSCs are isolated from a segment of small intestine from *Wnt1::Cre;R26iDTR* mice in which focal colonic aganglionosis is simultaneously created by diphtheria toxin injection. Autologous ENSCs are isolated, expanded, labeled with lentiviral-GFP, and transplanted into the aganglionic segment in vivo. ENSCs differentiate into neurons and glia, cluster to form neo-ganglia, and restore colonic contractile activity as shown by electrical field stimulation and optogenetics. Using a non-lethal model of colonic aganglionosis, our results demonstrate the potential of autologous ENSC therapy to improve functional outcomes in neurointestinal disease, laying the groundwork for clinical application of this regenerative cell-based approach.

The enteric nervous system (ENS) is an extensive network of neurons and glia within the wall of the gastrointestinal (GI) tract that regulates a number of GI functions, including motility[1]. Enteric neuropathies, in which enteric neurons are abnormal or absent, cause significant morbidity. Hirschsprung disease (HSCR), for example, is a congenital enteric neuropathy affecting 1 in 5000 children and characterized by the absence of ganglion cells along variable lengths of distal bowel due to failure of neural crest-derived precursors to complete their colonization of the developing intestine. The aganglionic intestine is functionally obstructed and treatment involves surgical resection of the aganglionic segment. While surgery is life-saving, many children have persistent GI problems, including constipation, faecal incontinence, and enterocolitis[2,3]. This morbidity has multiple causes, including the need for proctectomy to remove the aganglionic rectum, operative injury to sphincters and pelvic nerves, and residual aganglionosis or dysganglionosis in the remaining bowel[3]. Enteric neuronal cell transplantation represents a potential novel, curative therapy for HSCR that could mitigate many of these risks by repopulating the aganglionic distal bowel with neurons and thereby eliminate the need for proctectomy, one of the major causes of morbidity following pull-through surgery for HSCR[4].

Enteric neuronal stem/progenitor cells (ENSCs) can be isolated from the GI tract of rodent[5–8] and human[9–12] or derived from human pluripotent stem cells (PSCs)[13–17]. Following transplantation into the colon of postnatal rodents, ENSCs proliferate, migrate, and form clusters of differentiated neurons and glia cells[5,7,8]. Improvement in colonic motility has been demonstrated by transplantation of ENSCs into the colon of an animal model of enteric neuropathy[8] and PSC-derived enteric neuronal progenitors improved survival in a mouse model of HSCR[16,17]. While these results confirm the potential of ENSC transplantation for the treatment of colonic aganglionosis, the use of autologous ENSCs as the cell source has not previously been explored. This approach could accelerate the clinical application of cell replacement therapy by eliminating immunologic and ethical concerns raised by other stem cell sources[18].

In this study, we utilized a non-lethal mouse model of colonic aganglionosis and transplanted autologous ENSCs obtained from the

[1]Department of Pediatric Surgery, Massachusetts General Hospital, Harvard Medical School, Boston, MA, USA. [2]Department of Pediatric Surgery, The second affiliated hospital of Xi'an Jiaotong University, Xi'an, Shaanxi, China. [3]Drug Discovery Laboratory, Wakunaga Pharmaceutical Co., Ltd., Hiroshima, Japan. [4]Stem Cells and Regenerative Medicine, UCL Great Ormond Street Institute of Child Health, London, UK. [5]These authors contributed equally: Weikang Pan, Ahmed A. Rahman. ✉e-mail: rhotta@mgh.harvard.edu

same animal to the aganglionic segment. Our findings support the feasibility of isolating and transplanting autologous ENSCs and their ability to normalize contractile activity in the aganglionic smooth muscle, demonstrating the potential of this innovative approach and bringing cell therapy for enteric neuropathies closer to clinical application.

## Results

### Isolation, expansion, and labelling of autologous ENSCs

To isolate autologous ENSCs, a midline laparotomy was performed on 2–3-month-old *Wnt1::Cre;R26-iDTR* (Wnt1-iDTR) mice (Fig. 1A, Day 0, Laparotomy #1). A short segment of the small intestine was removed and an end-to-end anastomosis performed using 9–0 PDS suture (Fig. 1B, arrows and Fig. 1C). At the time of laparotomy, the colon was exposed and 4 μL of diphtheria toxin (DT) injected into the mid-colon wall to create focal colonic aganglionosis, as previously described[19] (Fig. 1A, Day 0, Laparotomy #1). Mice were recovered from anesthesia and maintained on a liquid diet through postoperative day 7 and returned to normal chow on day 8. Cells obtained from dissociated intestinal tissue were cultured to form enteric neurospheres (Fig. 1A, Days 0–7, Primary culture, and Fig. 1D), as described[6]. On day #7 of culture, ENSCs were transduced with a lentiviral vector expressing GFP on a CMV promoter[20] (Fig. 1A, Days 7–10, Viral transduction) to allow identification after transplantation, followed by an additional 4 days in culture (Fig. 1A, Days 10–14, Secondary culture; Fig. 1D'). Small intestinal resections measured 1.5 to 4.0 cm (mean 2.6 ± 0.7 cm, n = 12), and the length of resection correlated linearly with the number of neurospheres generated. An average of 477 ± 123 neurospheres (n = 12) were generated per cm of small intestine (Fig. 1E, $r^2 = 0.46$ and $p = 0.0148$).

### Successful engraftment, migration, and differentiation of transplanted autologous ENSCs

Neurospheres contain ENSCs that express P75 (Fig. 2A), glial cells (Fig. 2A', GFAP; Fig. 2B', S100), and neurons (Fig. 2B, Tuj1). Following enzymatic dissociation of neurospheres, ENSCs were cultured on fibronectin-coated glass chambers, where they differentiate into neurons (Fig. 2C, HU) and glial cells (Fig. 2C', GFAP), including a subpopulation of neurons (Fig. 2D, Tuj1, arrow and arrowhead) immunoreactive for neuronal nitric oxide synthase (nNOS) (Fig. 2D',

arrow). To determine whether autologous ENSCs can survive and differentiate within the aganglionic gut environment in vivo, a second laparotomy was performed on the same mice from which ENSCs had been isolated (Fig. 1A, Day 14, Laparotomy #2). The segment of mid-colon where DT had been injected was identified based on the presence of India ink, and 60–80 GFP+ autologous neurospheres were microinjected into the gut wall (Fig. 2E, arrowheads). Two weeks later (Fig. 1A, Days 28), the colon was removed and wholemount immunofluorescence identified transplanted ENSCs by GFP expression within the aganglionic area (Fig. 2F, dotted line demarcates focal aganglionosis). The pan-neuronal marker, Tuj1, confirmed neuronal differentiation of autologous ENSCs (Fig. 2F, F', arrowhead). Transplanted ENSCs were observed in the adjacent unablated, ganglionated gut, where they formed cell clusters containing both transplant-derived neurons (Fig. 2G, arrow) and endogenous enteric neurons (Fig. 2G, arrowhead).

### Autologous ENSCs form "neo ganglia" that contain enteric neuron subtypes and extensive neural fibre projections

Further examination of recipient mouse colon was performed immunohistochemically (Fig. 3A). Surface areas of ENS-ablated colonic segments were measured based on anti-HU immunostaining, showing that aganglionic areas were 57.1 ± 8.7 mm² at 5 weeks post DT injection (Fig. 3A, area marked by dotted line, n = 3) and 52.5 ± 5.0 mm² at 10 weeks following DT injection (n = 3). The mean cell coverage at 3 and 8 weeks post-transplant was 6.9 ± 1.1 mm² at 3 weeks and 6.1 ± 1.1 mm² at 8 weeks post-transplant, respectively (n = 3 of each). We also quantified the length of fibres projecting from transplanted ENSCs, which measured 4.3 ± 0.5 mm and 4.1 ± 0.5 mm at 3 and 8 weeks post-transplant, respectively (n = 3 of each). To evaluate the extent of ENSC migration across the gut wall, we performed z-series imaging using confocal microscopy. 3D reconstructions were made from z-stack images of double stained recipient colon using GFP and calretinin, a marker of excitatory enteric cholinergic neurons (Fig. 3B). Images showed the majority of extensively projecting fibers originated from transplanted ENSCs expressing calretinin. Z-stack images were also processed as maximum intensity projections (Fig. 3C) along different layers of the gut wall; at 25–55 μm (Fig. 3C'), 80–110 μm (Fig. 3C''), and 135–155 μm in depth (Fig. 3C'''). All images from different

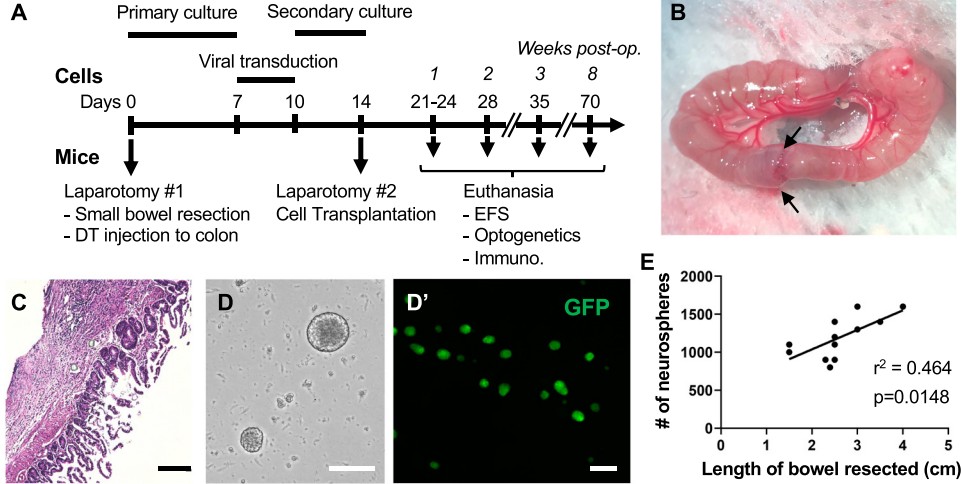

**Fig. 1 | Experimental overview of autologous ENSC transplantation. A** Timeline of experiments for isolating and expanding intestinal ENSCs from adult Wnt1-iDTR mice followed by autologous transplantation to diphtheria toxin-induced aganglionic colon. Immunohistochemical and functional assays were performed 1–8 weeks following cell transplantation. **B** End-to-end small intestinal anastomosis was performed (arrows) following segmental bowel resection. **C** Histologic examination confirmed tissue repair at the anastomosis one week after surgery. **D** ENSCs were expanded as neurospheres and labelled with lentiviral-GFP (**D'**). **E** Scatter plot and corresponding regression line show a linear relationship between length of bowel resected and number of neurospheres generated. DT, diphtheria toxin; EFS, electrical field stimulation. Scale bars 100 μm (**C**, **D'**), 200 μm (**D**).

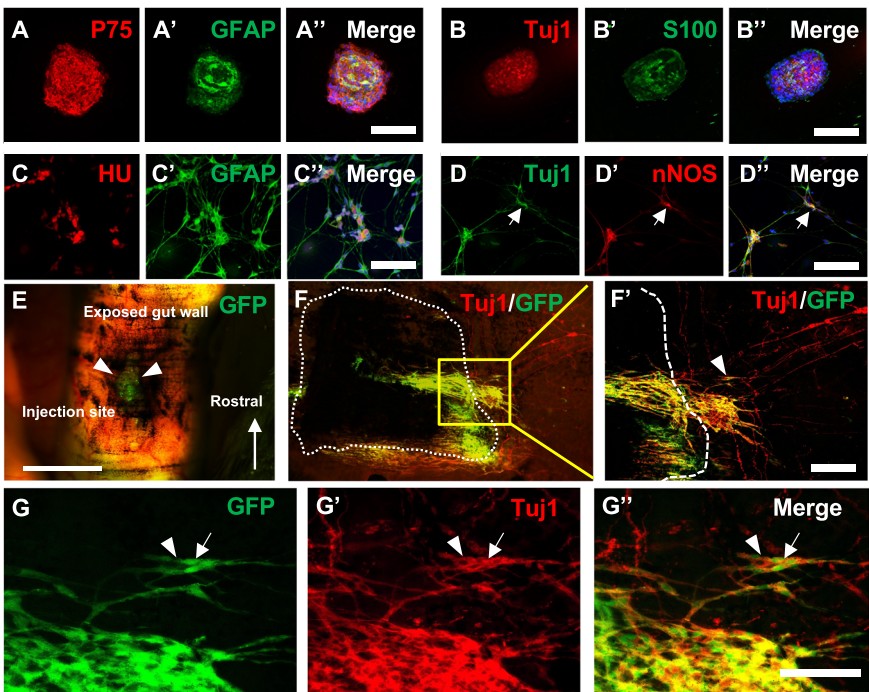

**Fig. 2 | Characterization of autologous ENSCs before and after transplantation.** Characterization of autologous ENSCs using markers for neural crest cells (**A**, P75), glial cells (GFAP in **A′**; S100 in **B′**), and neurons (**B**, Tuj1). Dissociated neurospheres were grown on fibronectin-coated cover slips and neuroglial differentiation confirmed (**C**). A subpopulation of the cells gave rise to nNOS-expressing neurons (**D**, arrows). ENSCs were transplanted to the mid-colon of DT-treated Wnt1-iDTR mice, and GFP-labelled autologous ENSCs were seen at the injection site (**E**, arrowheads). Two weeks later, successful engraftment, migration, and neurofibre extension of transplanted cells was seen (**F**), with differentiation into Tuj1-expressing neurons (**F**, box magnified in **F′**, where arrowhead shows GFP/Tuj1 co-expression). (**G–G″**) High power image of the transplant site shows integration of endogenous (GFP-; arrowhead) and transplant-derived (GFP + ; arrow) neurons. Scale bars, 100 μm (**A–A″**, **B–B″**, **F**, **F′**), 200 μm (**C–C″**, **D–D″**, **G–G″**), 500 μm (**E**).

layers captured GFP + /calretinin+ neural fibres, suggesting calretinin+ transplanted ENSC-derived neural fibres penetrate across and innervate the colonic muscle layers (Fig. 3B, and 3C–C‴). Subpopulations of transplanted ENSCs formed cell clusters, resembling enteric ganglia, which we termed "neo ganglia" (Fig. 3A and 3C″, arrows). High-power images of transplanted sites showed that neo ganglia contain calretinin (Fig. 3D–D‴) or nNOS (Fig. 3E–E‴) expressing enteric neuron subtypes and enteric glial cells (Fig. 3F–F″, S100β). We also examined the ENS composition of neo ganglia using immunohistochemistry, demonstrating 84.8 ± 3.2% (*n* = 3) of ENSC-derived neurons within the neo ganglia were nNOS neurons at 3 weeks post-transplant, whereas only 15.1 ± 6.4% (*n* = 3) were calretinin positive (Fig. 3G). Statistical comparison to the ENS composition in wildtype mouse colon (Fig. 3H, I) demonstrated significant differences in both subtypes (31.6 ± 2.8% nNOS neurons in control, *n* = 3 and 47.9 ± 3.7% calretinin neurons in control, *n* = 3, ***$p$ < 0.001, *$p$ < 0.05 in Fig. 3G). The proportion of calretinin+ neurons was increased at 8 weeks post-transplant (Figs. 3G and J–J″) and nNOS/calretinin composition appeared to shift towards that in control over time (70.2 ± 7.5% nNOS at 8 weeks, *n* = 3 and 40.3 ± 6.2% calretinin at 8 weeks, *n* = 3, Fig. 3G); however, a predominance of nNOS neurons remained (**$p$ < 0.01 in Fig. 3G).

**Autologous-derived ENSCs restore neuromuscular activity in aganglionic colon**
Organ bath studies were performed 3 and 8 weeks following cell transplantation (Fig. 1A, Days 35 and 70). Colonic smooth muscle preparations obtained from control mice showed non-rhythmic, frequent, low-amplitude contractions (Fig. 4A, left panel, "Spontaneous" in Control trace), whereas DT-induced aganglionic colon exhibited high-magnitude rhythmic contractions during baseline recordings (Fig. 4A, left panel, "Spontaneous" in DT-AG trace). To determine the

neuronal contribution to these abnormal contractile activities, the sodium channel blocker, tetrodotoxin (TTX, 0.5 μM), was added to the organ bath. The high-amplitude contractile activity was not blocked by TTX (Fig. 4B, DT-AG trace), suggesting a myogenic origin underlying this activity.

Interestingly, DT-AG colon where cells were transplanted showed no spontaneous high-amplitude rhythmic contractions (Fig. 4A, "Spontaneous" in DT-AG + Cells). However, the addition of TTX led to more prominent contractile activity (Fig. 4B, "TTX response" in DT-AG + Cell trace), similar to that seen in TTX-treated ganglionated colon obtained from control mice (Fig. 4B, "TTX response" in Control trace). These spontaneous activities were quantitatively analyzed by measuring area under the curve (AUC) at random 60 second time periods, and demonstrated significantly larger areas in the aganglionic colon (Fig. 4C, 78 ± 21 g.s in DT-AG, *n* = 4) compared to the ganglionic colon (Fig. 4C, 9 ± 3 g.s in Control, *n* = 4; *$p$ < 0.001). Importantly, these alterations in the aganglionic tissues were partially restored by cell transplantation (Fig. 4C, 28 ± 6 g.s in DT-AG + Cells, *n* = 4; *$p$ < 0.01).

Next, we evaluated the contractile activities of colonic smooth muscle using electrical field stimulation (EFS). A robust contractile response was induced by EFS in colonic smooth muscle from control mice that received DT injection (Fig. 4A, right panel, "EFS response" in Control; Fig. 4D, 3.11 ± 0.13 gm in Control). In contrast, no response was seen in colonic smooth muscle from Wnt1-iDTR mice treated with DT (Fig. 4A, right panel, "EFS response" in DT-AG; Fig. 4D, 0.10 ± 0.09 gm in DT-AG, *n* = 4). This dramatic reduction in EFS-evoked muscle contractility of DT-induced aganglionic muscle was significantly improved 3 weeks after ENSC transplantation (Fig. 4A, right panel, "EFS response" in DT-AG + Cells; Fig. 4D, 0.10 ± 0.09 gm in DT-AG vs 1.45 ± 0.23 gm in DT-AG + Cells (3 weeks), *n* = 4 each, ***$p$ < 0.001). Interestingly, the effect of ENSC transplantation on

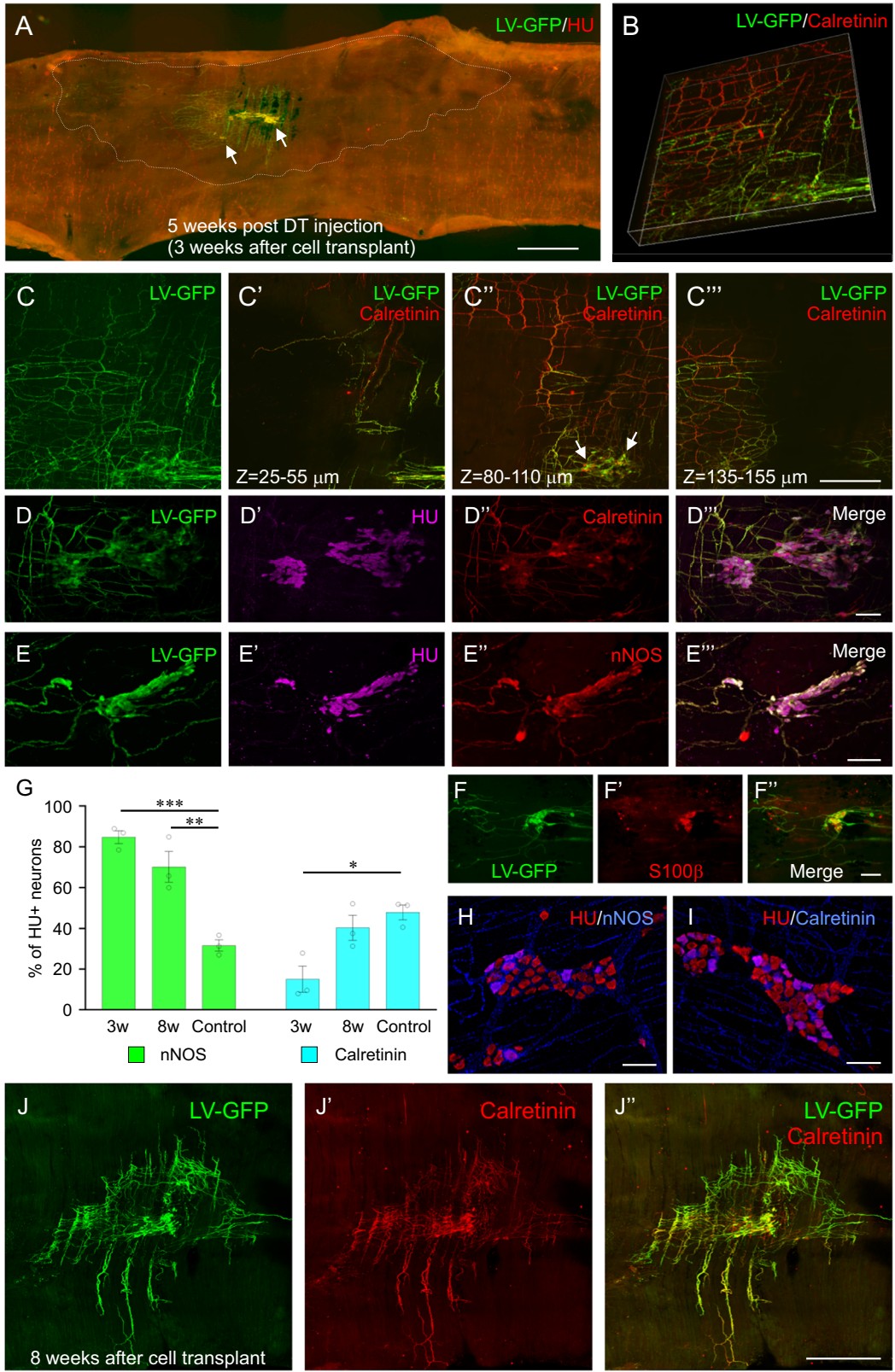

smooth muscle contraction improved over time, with normalization of smooth muscle contractility at 8 weeks post-transplant (Fig. 4D, 1.45 ± 0.23 gm in DT-AG + Cells at 3 weeks vs 2.95 ± 0.39 gm in DT-AG + Cells at 8 weeks, *n* = 4 each, **p < 0.01). To confirm these effects are neurally mediated, TTX (0.5 μM) was added to the organ bath. This significantly reduced the contractile response to EFS in the control colon (Fig. 4E, 34.5 ± 8.1%, ***p < 0.01), as expected. Importantly,

contractility in ENSC transplanted colon was also significantly reduced (Fig. 4E, 7.7 ± 10.4%, ***p < 0.01), demonstrating that the restoration of muscle contraction in aganglionic colon is mediated by the transplanted neurons. To confirm that the motor apparatus is intact following neuronal ablation, and that the differences observed are not due to smooth muscle abnormalities, we tested the cholinergic response of the colonic smooth muscle preparations to acetylcholine

**Fig. 3 | Wholemount immunohistochemistry of recipient colon and characterization of "neo ganglia".** Immunohistochemical examination of wholemount preparations of recipient colon showed successful induction of colonic aganglionosis (**A**, dotted area) and engraftment of transplanted cells with extensive fiber projections 3 weeks post cell transplant (**A**). 3D reconstructed confocal stack (**B**) was evaluated using maximum intensity projections (**C**) performed at different tissue depths; 25–55 μm (**C′**), 80–110 μm (**C″**), and 135–155 μm (**C‴**). High power images of the transplant site showed "neo ganglia" containing calretinin (**D–D‴**) or nNOS (**E–E‴**) expressing enteric neuron subtypes and S100β positive enteric glial cells (**F–F″**). Quantitative analysis characterized neo ganglia by enteric neuron subtypes (**G**) compared to control (**H**, nNOS and **I**, Calretinin). $n = 3$ for each group. ***$p < 0.001$, **$p < 0.01$, *$p < 0.05$ by 2 way ANOVA. Successful survival of transplanted autologous ENSCs 8 weeks following transplantation (**J, J″**) with extensive calretinin immunoreactive fiber projections (**J′, J″**). Error bars represent mean ± SEM in all panels. Scale bars, 50 m (**H, I**) 100 μm (**D–D″, E–E‴**, and **F–F″**), 500 μm (**C–C‴**), 1 mm (**J–J″**), 2 mm (**A**).

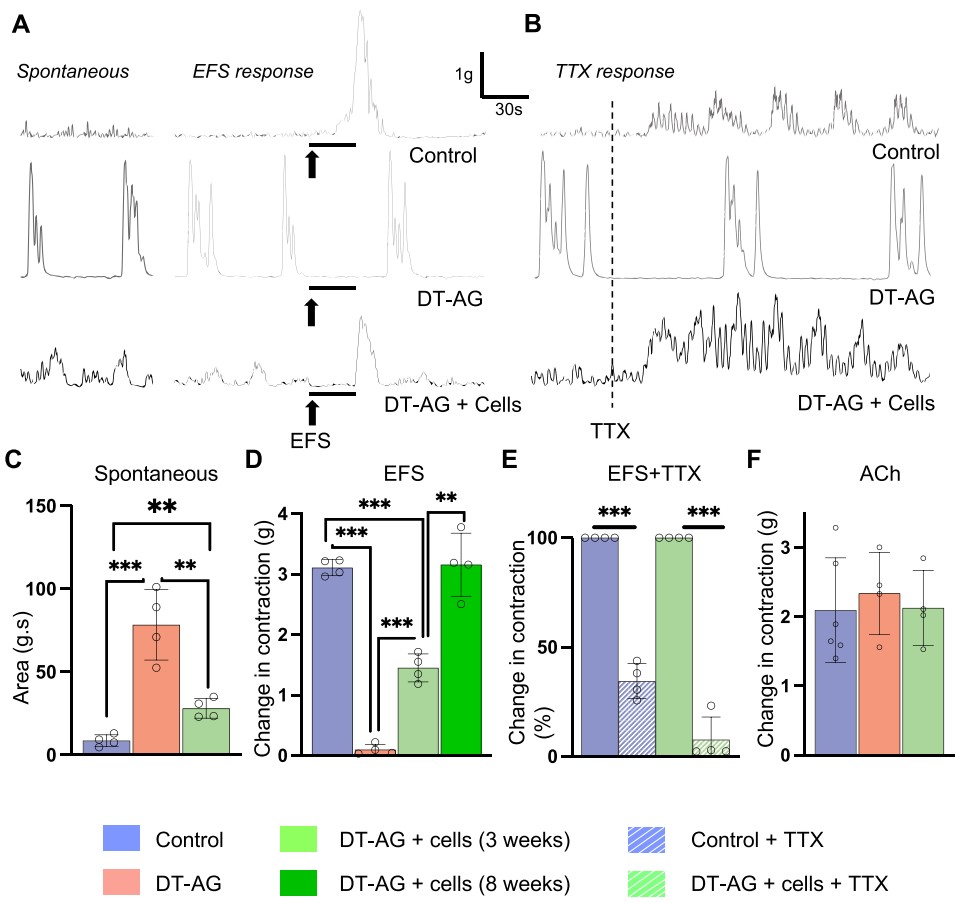

**Fig. 4 | Functional recovery of aganglionic smooth muscle post-cell transplantation. A** Representative traces of spontaneous smooth muscle contraction (left panel) and contraction responses to EFS (right panel). **B** Representative traces of smooth muscle activities in the presence of TTX. **C** Quantification of spontaneous muscle contraction without any stimulus (area under curve). $n = 4$ of each group. ***$p < 0.001$; **$p < 0.01$ by Student's $t$-test. **D** EFS-induced muscle contraction 3 and 8 weeks after cell transplantation (maximum effects are shown as absolute change from baseline values). $n = 4$ of each group. ***$p < 0.001$, **$p < 0.01$ by Student's $t$-test. **E** EFS response was significantly reduced by TTX. $n = 4$ of each group. ***$p < 0.001$ by Student's $t$-test. **F** All groups contract in response to ACh, confirming intact colonic contractile function. $n = 4$ of each group. Error bars represent mean ± SEM in all panels. DT-AG, diphtheria toxin-induced aganglionosis; EFS, electrical field stimulation; TTX, tetrodotoxin.

(ACh). All three groups demonstrated similar contractile responses in response to this excitatory neurotransmitter (Fig. 4F, 2.1 ± 0.76 gm in Control, $n = 6$; 2.34 ± 0.73 gm in DT-AG, $n = 4$; 2.13 ± 0.54 gm in DT-AG + Cells, $n = 4$; ns).

### Optogenetic activation of transplanted ENSCs confirms neuronal activity of transplanted ENSCs

To confirm that EFS-induced responses were due to the transplant-derived cells, we transduced ENSCs with lentivirus encoding the light-sensitive ion channel, channelrhodopsin-2 (ChR2) (Fig. 1A, Days 0–10, Viral transduction), and transplanted these cells to aganglionic colon as above. Three weeks later, organ bath studies were performed. Colonic rings obtained from DT-induced aganglionic colon, but without cell transplantation, displayed no contraction in response to blue

light stimulation (BLS) (Fig. 1A, DT-AG). In contrast, BLS induced a significant contractile response in colons receiving ChR2+ cells (Fig. 5A, DT-AG + ChR2 cells). Quantitative comparison demonstrated a significant increase in the amplitude of muscle contraction (Fig. 5B, 0.09 ± 0.06 gm in DT-AG vs 1.17 ± 0.36 gm in DT-AG + ChR2, **$p < 0.01$). The addition of ACh (Fig. 5C) or KCl (Fig. 5D) confirmed intact motor response in both transplanted and non-transplanted preparations.

## Discussion

This is the first study to demonstrate successful transplantation of autologous ENSCs in an in vivo model of neurointestinal disease. Importantly, we show that ENSC transplantation restores neuromuscular function following experimentally induced colonic aganglionosis. Our findings establish methods for the isolation and expansion

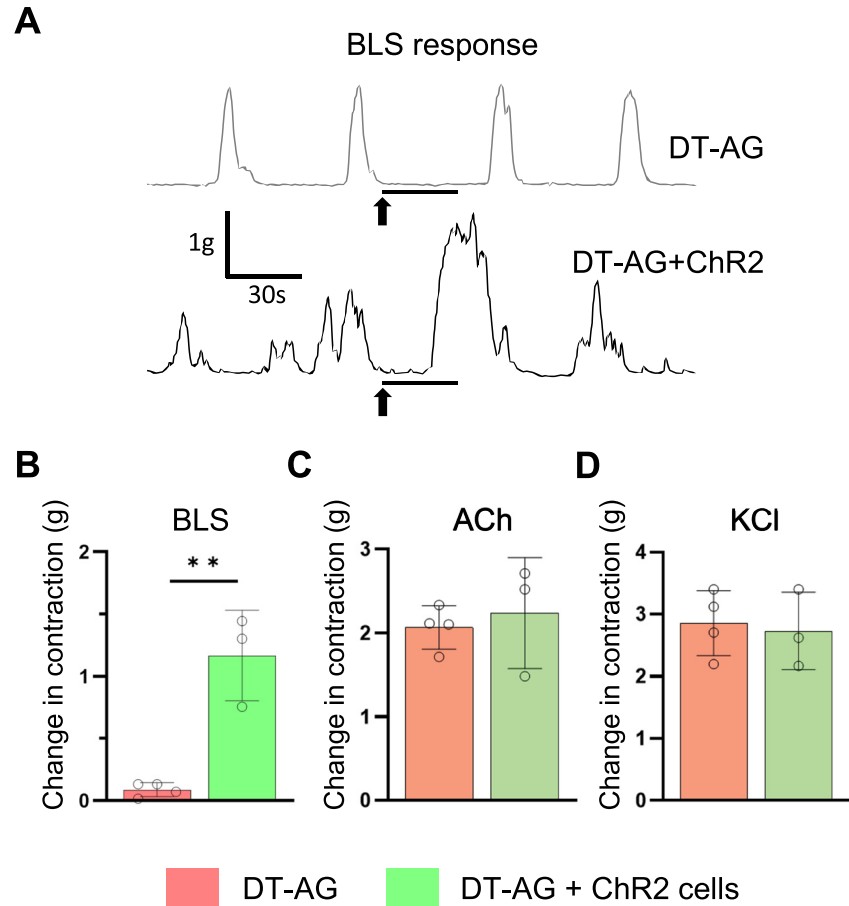

**Fig. 5 | Optogenetics confirms neuromuscular connectivity between ENSCs and recipient aganglionic colon. A, B** Representative traces of muscle contraction to blue light stimulation (BLS). Arrows indicate the time of stimulation. DT-AG colon showed no response, while BLS induced muscle contraction in colon transplanted with ENSCs expressing lentiviral-ChR2. $n = 4$ of each group. **$p < 0.01$ by Student's t- test. ACh (**C**) and KCl (**D**) induced colonic smooth muscle contraction, consistent with normal colonic motor function. $n = 4$ of each group. Error bars represent mean ± SD in all panels. ACh, acetylcholine; BLS, blue light stimulation; ChR2, channelrhodopsin 2; DT-AG, diphtheria toxin-induced aganglionosis.

of autologous ENSCs, and their successful transplantation into recipient colon. ENSCs survived within the aganglionic environment for at least 8 weeks, differentiated into neurons and glia, formed ganglia-like clusters, and rescued colonic contractile activity.

Testing the feasibility, safety and efficacy of autologous cell therapy using animal models is a prerequisite before advancing to clinical trials. Due to the limited lifespan of rodent models of Hirschsprung disease, animals such as the Ednrb-deficient mouse or rat are not amenable to long-term follow-up beyond a few weeks postcell transplant. Although we have previously shown that transplanted ENSCs can engraft and differentiate into neurons and glial cells within the aganglionic gut environment ten days after transplantation in vivo[21], to date no studies have examined the transplanted cells beyond 2 weeks following transplantation. An elegant surgical rescue approach has recently been developed to prolong the life of Ednrb null rat pups as recipients for cell transplant by creating a stoma to allow the normally innervated bowel to empty and thus bypass the constricted aganglionic segment[22]. However, in this study cells were not transplanted into the colon of these rescued animals.

To overcome the problem having an optimum disease model with good survival allowing longer-term follow-up, we recently established a non-lethal model of colonic aganglionosis using a DT-mediated neural crest cell ablation strategy that leads to focal aganglionosis with associated colonic motor dysfunction, but without lethality or other systemic consequences[19]. In this model, the fate of transplanted cells can be followed for longer periods, and by allowing time for cells to

establish functional neuromuscular connections, the efficacy of autologous cell therapy for aganglionosis can be tested. Using these DT-treated mice, we recently showed that Schwann cell-derived neuronal progenitors can improve muscle contractility of the aganglionic colon[23,24]. Consistent with our previous work, in this current study, we observed an improvement in contractile activity at 3 weeks following cell transplantation. Importantly, this improvement progressed over time and at 8 weeks post cell transplantation we observed full normalization of contractility.

Along with the improvement in gut contractile activity following cell transplantation, we also made some interesting observations regarding the nature and origin of the contractile events. In the DT-treated aganglionic colon, there was a consistent increase in spontaneous contractile events, which displayed regular, rhythmic patterns. Sustained inhibition of colonic motility provided by tonic inhibitory neuronal input to the colonic smooth muscle is required to absorb liquid from luminal contents and store faeces in the colon. We observed this phenomenon in the smooth muscle preparations obtained from control mice as reflected by the presence of non-rhythmic, frequent, low-amplitude contractions. Due to the absence of neurons, including inhibitory neurons, in the DT-induced aganglionic colon, we observed high-magnitude rhythmic contractions in preparations from this region. However, the aganglionic colonic muscle no longer showed obvious high-amplitude rhythmic contractions after cell transplantation, likely due to inhibitory innervation provided by the transplanted ENSCs. In order to determine the neuronal

contribution to the rhythmic activities in the DT-induced aganglionic smooth muscle, the voltage-gated sodium channel blocker, TTX was added to the bath in contractile studies. In previous experiments, TTX induced higher muscular contractions[25–27], which are believed to be due to the removal of tonic inhibitory neural regulation that suppresses myogenic contractility[28,29]. Similarly, in our studies, the addition of TTX resulted in a contraction in the colon of control and cell-transplanted mice likely by blocking inhibitory neural input to smooth muscle cells, but no changes were observed in the myogenic contractions of the DT-AG colon (Fig. 4B). These findings confirm that colonic tissues that are the recipients of cell transplants have newly developed functional neurons that communicate with smooth muscle cells, and the lack of neurons in the aganglionic colon exposes underlying myogenic activity that is not altered by TTX.

While the focal ablation approach has the significant advantage of being non-lethal, allowing longer-term testing of the ability of autologous neural stem cells to form ENS and confer function onto denervated smooth muscle, it is not a mimic of naturally occurring aganglionosis. The Wnt1::Cre; R26-iDTR transgenic strain used in the study that allows the generation of aganglionosis by focal injection of DT, does not carry any of the genetic defects that have been reported either in HSCR rodent models or in HSCR patients. The local aganglionic gut region is generated in a targeted way in postnatal mice and does not phenocopy the failure of enteric neural progenitors to colonize the gut early in development due to defects in intrinsic cell migration, proliferation or differentiation. The absence of enteric neurons in the terminal colon during the development of rodents and HSCR patients has been reported to have a number of implications, to varying extents, on the local gut environment, including on the smooth muscle, other cell types such as interstitial cells of Cajal (ICC), the extracellular matrix, the microbiome, and on extrinsic nerve fibres[30–35]. This aspect of the model, and a number of other caveats, should be born in mind when considering our studies using the focal ablation approach.

As mentioned above, the autologous stem cells used for transplant were obtained from the same Wnt1::Cre; R26-iDTR mice into which they are subsequently transplanted. Thus, the autologous cells used in this study do not have any of the genetic defects that have been reported either in HSCR rodent models or in HSCR patients[15]. One would therefore not expect the autologous cells to be adversely affected by any underlying disease-related genetic abnormalities that are characteristic of aganglionosis, but to behave normally until exposed to DT. This raises the question - if transplanted cells in the model survive, form ENS and restore contractile activity, would transplanted autologous cells from rodent models of HSCR or HSCR patients behave in the same way and restore gut function? This issue is challenging to address for the reasons concerning the survival of HSRC genetic animal models as mentioned above, and it is unlikely that autologous cells will be transplanted into HSCR patients without compelling proof of concept supporting data in place. As such, while not a disease-mimicking cell type, the transplant of ENSCs from Wnt1::Cre; R26-iDTR into the focally generated aganglionic region is, nevertheless, a valid assessment of the ability of autologous neural stem cells to engraft, migrate, extend neurites, become integrated with the neuromuscular circuitry, and confer function onto denervated smooth muscle.

HSCR is regarded as a neurocristopathy where the primary defect resides in the neural crest cells that give rise to the ENS. What the cellular defects in the enteric neural crest cells are, how do they result in the failure to colonize the entire length of the gut, and how do any coding or non-coding genetic mutations in HSCR patients relates to cellular and molecular defects in these cells, has been intensely studied, but remains complex and not well understood. Regardless of any genetic and cellular defects, in the majority of HSCR patients (those with short segment disease, ~70% +), most of the gut is colonized by

enteric progenitors, with only the very terminal region of the colon remaining aganglionic. This suggests that in HSCR, enteric neural crest cells have sufficient migration/proliferation/differentiation capacity to form most of the ENS. Consequently, ENSCs harvested from the "normoganglionic" region of HSCR patient bowel likely have sufficient ENS-forming capability to be used for autologous cell therapy. We have published data showing that ENSCs obtained from the Ednrb-/-mouse model of HSCR have capacity for self-renewal and neuronal differentiation and that the extent of proliferation and differentiation of these Ednrb-/- derived cells is not significantly different from that of cells derived from wild type littermates[21]. Our studies, using autologous ENSCs from a non-genetic HSCR model, although not perfect, help to advance the overall body of work on autologous stem cells and progress the field towards the therapeutic use of autologous ENSCs for HSCR.

We and others have previously shown successful isolation and expansion of ENSCs from mouse and human intestine and demonstrated their ability to engraft and differentiate into neurons and glial cells following allotransplantation or xenotransplantation into mouse models of enteric neuropathies[8,10,16,21,36]. These observations serve as proof of concept that cell therapy can be a promising therapeutic approach to replace missing or damaged enteric neurons to restore GI function[4,37]. However, the immunogenicity associated with utilizing non-autologous ENSCs remains a major concern[12,38], as does the ethical challenge associated with using embryonic stem cells or the long-term safety risks associated with PSCs. While autologous-derived ENSCs hold significant advantages over the use of allogeneic cells[38,39], prior work has not investigated the feasibility or efficacy of this approach for the reasons mentioned above. A potential concern with autologous ENSCs obtained from a patient with an enteric neuropathy is whether those cells are capable of restoring gut function. Several studies have, encouragingly, demonstrated successful derivation of ENSCs from human patients with enteric neuropathies[12,15] and their engraftment, migration, and neuroglial differentiation within gut explants from HSCR patient tissue[12]. In our current study, the repopulation of the focal aganglionic region by transplanted ENSCs allowed us to demonstrate the ability of these cells to restore EFS-induced smooth muscle contractile events, indicating the establishment of functional ENS networks and resulting restoration of neuromuscular function. However, we were not able to test for changes in peristaltic activity, the ultimate goal of cell transplant in the human therapeutic setting. The focal ENS ablation model of aganglionosis was initially established in our laboratory[19]. We demonstrated that global ablation of Wnt1-iDTR expressing ENS cells by systemic injection of DT resulted in the loss of ENS cells and colonic dysmotility as shown by an absence of colonic migrating motor complexes (CMMCs), which underlie the propulsive activity of peristalsis, using spatiotemporal mapping. However, when GI transit and colonic motility was examined in mice with focal ablation of ENS using local injection of DT into the colon, we did not observe changes in gut transit times, neither were there changes in CMMC frequency, length, or velocity in DT-treated mice as compared with sham-operated controls[19]. Thus, although the focal ablation model provides a significant advance by avoiding animal mortality, CMMCs, and colonic peristaltic activity are not affected, likely due to the relatively small size of the aganglionic region generated. Therefore, in the current study, functional studies focused on neuromuscular contractile measurements using rings of gut tissue from the transplanted region analyzed using organ bath pharmacology. We did not assess for any potential changes in CMMCs in the focally ablated region postcell transplant as we have already shown that there are no CMMC defects to attempt to rescue in this model.

Overall, this study demonstrates the feasibility of an autologous strategy to treat intestinal aganglionosis and brings the field a step closer to clinical application of cell therapy for enteric neuropathies.

## Methods

### Animals

All animal protocols were approved by the Institutional Animal Care and Use Committee at Massachusetts General Hospital (Protocols #2009N000239 and #2013N000115). All methods were carried out in accordance with relevant guidelines and regulations. The reporting in the manuscript follows the recommendations in the ARRIVE guidelines.

The following mouse lines were obtained from Jackson Laboratory (Bar Harbour, ME, USA): *Wnt1::Cre* mice (Stock #009107) were crossed with *R26-iDTR* mice (Stock #007900) to obtain *Wnt1::Cre; R26-iDTR* (Wnt1-iDTR) mice. R26-iDTR mice were used as controls. All mice were used at 2–3 months of age and included both sexes.

### Isolation and culture of enteric neural stem/progenitor cells (ENSCs)

Longitudinal muscle-myenteric plexus (LMMP) layers were dissected from the mouse intestinal tissues, minced, and dissociated with Dispase (250 µg/mL; StemCell Technologies, Vancouver, BC) and collagenase XI (1 mg/mL; Sigma Aldrich, St. Louis, MO) at 37 °C for 25 s. A 40 µm cell strainer (Corning Inc, Corning, NY) was used to collect a single cell suspension. Cells were cultured for 7–10 days in mouse proliferation media, consisting of Neurocult Mouse Basal Medium (StemCell Technologies) supplemented with 10% Neurocult Mouse Proliferation Supplement (StemCell Technologies), 20 ng/mL epidermal growth factor (StemCell Technologies), 10 ng/mL basic fibroblast growth factor (StemCell Technologies), and 0.0002% Heparin (StemCell Technologies), and allowed to form primary cell aggregates (enteric neurospheres).

### Tissue and cell preparation and immunohistochemistry

Preparation of tissues and cells for immunohistochemistry was performed as previously described[6,19]. Cells, LMMP, and full-thickness gut samples were fixed in 4% paraformaldehyde. For cryosections, full-thickness gut samples were embedded in 15% sucrose at 4 °C overnight, and then in 15% sucrose with 7.5% gelatin at 37 °C for 1 h. The tissue was rapidly frozen at −50 °C in liquid nitrogen. Frozen sections were collected on glass slides at 12–14 µm thickness with a Leica CM3050 S cryostat (Leica, Buffalo Grove, IL). For immunohistochemistry, the samples were permeabilized with 0.1% Triton X-100 and blocked with 10% donkey serum for 30 ms.

Primary antibodies were diluted in 2% donkey serum, 0.01% Triton X-100 and included human anti-HU (Anna1, 1:16000, kindly gifted by Lennon lab), mouse anti-neuronal class III β-tubulin (Tuj1; 1:400; conjugated to Alexa Fluor 546, Invitrogen), goat anti-GFP (1:400, Rockland, Limerick, PA), rabbit anti-glial fibrillary acidic protein (GFAP, 1:200, Agilent/Dako, Denmark), rabbit anti-neuronal nitric oxide synthase (nNOS; 1:200, Thermo Fisher), rabbit anti-calretinin (1:200, Invitrogen), rabbit anti-p75 neurotrophin receptor (P75; 1:400; Promega, Madison, WI), and rabbit anti-S100 beta antibody (1:100, Abcam, US).

Secondary antibodies included donkey anti-human IgG (1:200, Alexa Fluor 488 and 647; Fisher Scientific Life Technologies), donkey anti-goat IgG (1:500; Alexa Fluor 488; Fisher Scientific Life Technologies), and donkey anti-rabbit IgG (1:500; Alexa Fluor 488 and 546; Fisher Scientific Life Technologies). Cell nuclei were identified with DAPI solution (Vector Labs, Burlingame, CA) and mounted with Aquapoly/mount (Fisher Scientific Polysciences Inc).

Images were taken using a Nikon A1R laser scanning confocal microscope (Nikon Instruments, Melville, NY), Nikon AXR confocal microscope (Nikon Instruments), or a Keyence BZX-700 All-In-One Microscopy (Keyence America Itasca, IL).

### Viral transduction

Lentivirus vector expressing firefly luciferase and GFP (LV-GFP), separated by an internal ribosomal entry site, under the control of the CMV promoter[20] with a titre of $2.3 \times 10^8$ to $1.0 \times 10^9$ IU/mL was prepared by the MGH Viral Vector Core Facility (Massachusetts General Hospital Neuroscience Centre, Charlestown, MA). Adeno-associated virus serotype type 9 (AAV9) expressing light-activated ion channel, channelrhodopsin-2 (ChR2) and fluorescent protein Venus under the CAG promotor (AAV9-CAG-ChR2-Venus) was purchased from Signa-Gen Laboratories (Frederick, MD, USA; Cat #SL100851, titer $>1 \times 10^{13}$ VG/mL). ENSCs were transduced with LV-GFP or AAV9-CAG-ChR2-Venus vectors at multiplicity of infection (MOI) in the range of 3–6[6,21] or 700–2000, respectively. Successful transduction was confirmed by GFP or Venus expression prior to cell transplantation.

### Segmental bowel resection

Solid food was replaced with a liquid diet[40] 2–3 days prior to the procedure. Mice were anesthetized with isoflurane inhalation. A 2 cm midline abdominal incision was made and a 1.5–4 cm segment of small intestine resected about 10 cm proximal to the caecum The resected segment of small intestine is used to prepare autologous ENSCs as described above. Small bowel anastomosis was performed using fine absorbable (9–0 PDS) interrupted sutures.

### Diphtheria toxin injection

At the time of the segmental small intestinal resection, the mid-colon of the same mouse was microinjected with 4 µl of 0.5 µg/ml diphtheria toxin (DT) to create a focal patch of aganglionosis in the colon[19]. India ink is included in the DT solution for easy identification of the aganglionic region.

### Cell transplantation to DT-induced aganglionic colon in vivo

One week after DT injection, mice were anesthetized and the same laparotomy incision was re-opened. A suspension of ENSCs was prepared containing about 10 neurospheres per microliter, and 5–10 µl was microinjected into the aganglionic colon in the area marked by India ink using a NanoFil™ microliter syringe (33 G, NF33BV-2, World Precision Instruments, FL, USA).

### Electrical field stimulation (EFS)

Experiments were performed using the standard organ bath technique as described previously[23,24]. Freshly excised colon was quickly placed in a Petri dish containing physiological Krebs solution. The colonic segment marked by Indian ink was cut into a 5 mm ring and mounted between two small metal hooks attached to force displacement transducers in a muscle strip myograph bath (Model 820 MS; Danish Myo Technology, Aarhus, Denmark) containing 7 ml of physiological Krebs solution (oxygenated with 95% O2 and 5% CO2) maintained at 37 °C. The rings were gently stretched to give a basal tension of 0.5 g and were equilibrated for 60 min in Krebs solution changed every 20 min. Colon segments were stimulated with pulse trains of 10–50 V for 30 s, with pulse duration of 300 µs, at a frequency of 5 Hz using a CS4+ constant voltage stimulator with Myo Pulse software (Danish Myo Technology, Aarhus, Denmark). Force contraction of the circular smooth muscle was recorded and analyzed using a Power Lab 16/35 data acquisition system (ADInstruments, NSW, Australia) and Lab Chart Pro Software v8.1.16 (ADInstruments). Acetylcholine (ACh, 100 µM, Sigma) was added to the organ bath to measure maximum contraction. Tissue viability and integrity were checked by eliciting contraction response to 60 mM KCl at the end of the study.

### Optogenetics

Segments of the transplanted colon were dissected and prepared as above. Blue light stimulation (BLS) was applied from a diode-pumped solid-state laser system (470 nm, 200 mW, Model number: MDL-III-470; OptoEngine, LLC, Midvale, UT). Trains of light pulses (20 ms pulse width, 20 mW/mm² light intensity, 10 Hz, 30 s train duration) were

focally directed on the serosal surface of the transplanted colon in the organ bath via a glass fiberoptic (200 μm diameter). Light intensity was assessed using Power and Energy Metre Interface (PM100USB, Thorlabs) and Standard Photodiode Power Sensor (S121C, Thorlabs).

## Data acquisition and analysis of organ bath studies

Baseline maximum value was obtained from 60 sec of data collected for 1 min prior to EFS or BLS. Maximum changes for contraction were taken from 60 sec of data starting at the time of the stimulus. These were expressed as absolute changes from baseline maximum values. EFS and BLS were repeated 3 times in 5 min intervals, with the maximum response calculated as a mean of 3 responses.

## Statistical Analysis

Data analyses were performed using Prism 9 (GraphPad software, Inc., La Jolla, CA, USA) and presented as mean ± standard deviation. Simple linear regression analysis was performed to determine the correlation between neurosphere number and length of intestinal resection. A one-way analysis of variance (ANOVA) was performed with a post-hoc Tukey's test for multiple comparisons. For all analyses, $P$ values < 0.05 were regarded as significant.

## Reporting summary

Further information on research design is available in the Nature Portfolio Reporting Summary linked to this article.

## Data availability

The datasets used and/or analysed during the current study available from the corresponding author on reasonable request. Source data are provided with this paper.

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

## Acknowledgements
This study was supported by the following funding sources: American Neurogastroenterology and Motility Society (A.R.); Shore Faculty Development Fellowship Award (A.R.); National Institutes of Health (R03HD100762, R.H.; R21HD106036, R.S.; R01DK119210, A.M.G.); Crohn's and Colitis Foundation (RS); Japan Society for the Promotion of Science (T.O.).

## Author contributions
Conceptualization, R.H., R.S., and A.M.G. methodology, R.H., W.P., T.O., K.O., and A.R. formal analysis, W.P., T.O., A.R., and R.S. investigation, R.H., W.P., T.O., K.O., and A.R. resources, R.H., A.J.B., and A.M.G. data curation, R.H., W.P., T.O., R.S., and A.R. writing original draught preparation, R.H., W.P., and A.R. writing review and editing, A.J.B., and A.M.G. visualization, R.H., W.P., R.S., and A.R. supervision, R.H., A.J.B., and A.M.G. funding acquisition, R.H., and A.M.G. C.Y.H., A.L., and A.K.

provided technical assists on various experiments. All authors have read and agreed to the published version of the manuscript.

## Competing interests
At the time of the study A.J.B. was an employee of Takeda Development Centre Americas and holds stock and/or stock options in Takeda. R.H., W.P., T.O., K.O., C.Y.H., A.L., A.K., A.R., R.S., and A.M.G. do not possess any competing interest.
