## [Peer Review File · Nature Communications]

Autologous cell transplantation for treatment of colorectal aganglionosis in miceREVIEWER COMMENTS

Reviewer #1 (Remarks to the Author):

This is an excellent paper that reports an important step on the road to devising a clinically useful treatment of enteric neuropathies with transplants of autologous cells. A major strength of the work is that the authors have devised a means of producing a limited degree of colonic aganglionosis that allows mice to survive long enough to permit an autologous graft of enteric neural crest-derived stem cells to be obtained, cultured, marked, transplanted, and enhance neuromuscular function in the region of bowel that was rendered aganglionic. Although the observations are not unexpected, the demonstration is striking in the elegance of the methods and the necessity of going from predictable to demonstrable. A weakness of the study, which the authors acknowledge at the end of the discussion, is that the lesion produced with diphtheria toxin injection into the colon of *Wnt1::Cre;R26-iDTR* (*Wnt1-iDTR*) mice, is both limited in size and highly artificial. The aganglionic region, while real, does not actually mimic any of the conditions that prevail, either in mice with a genetic cause of aganglionosis, or in humans with a genetic disorder, such as Hirschsprung disease (HSPR). In the authors' model, enteric neurons express DTR and thus die when exposed to DT. In the natural conditions, neurons and even crest derivatives may not be the only cells that are affected by the abnormality. The microenvironment of the aganglionic colon may itself be abnormal. Abnormalities of the extraganglionic matrix, for example, have been described in humans with HSPR and in murine models of the condition. The aganglionic region in HSPR and its murine models, furthermore, is not denervated. It is heavily innervated by thick nerve trunks that grow into the aganglionic bowel from more proximal ganglionic segments of the gut and from extrinsic ganglia. These nerves too may change the microenvironment, but they also illustrate the need for peristalsis that depends on the presence of a functional ENS. The authors' model, therefore, while aganglionic, is not a precise mimic of the naturally occurring aganglionosis of HSPR. The ability of autologous neural stem cells to survive and differentiate in the colon of *Wnt1::Cre;R26-iDTR* mice does not in any way suggest that the cells will do the same in a tissue in which endothelin or Ret signaling are abnormal.

A second perceived weakness of the authors' approach, again discussed only briefly at the end of the manuscript, is that the autologous stem cells from the small intestine of the *Wnt1::Cre;R26-iDTR* mice are themselves normal unless they encounter DT. Autologous stem cells from the small intestine of a human with HSPR or a mouse model of that condition, might not be normal and may be affected by the genetic abnormality at the root of the condition. Again, the authors' model does not mimic a natural disease. Because the transplanted cells in the model seem to survive and restore some function, one cannot conclude that similarly transplanted autologous cells will do the same in a patient or even in a mouse model of HSPR.

A final problem is that the authors have not tested peristaltic activity or propulsion. They have looked only at neuromuscular function. The authors have shown that destruction of colonic motor neurons removes cholinergic excitation of smooth muscle and that restoration of such neurons can restore cholinergic neural drive to the smooth muscle. These experiments are as elegant as the results are anticipated, but they do not speak to therapeutic efficacy. The authors maintain that they cannot

measure peristalsis in the colon after injecting DT into Wnt1::Cre;R26-iDTR mice because the lesion is too small; however, they do not say whether or not they tried to do evaluate peristalsis. Many labs have measured colonic migrating motor complexes (CMMCs) ex-vivo in mice, which are essentially peristaltic reflexes driven by the ENS. The authors report that electrically driven contractility of the muscle is absent in aganglionic bowel of their model. Are CMMCs present? Can they be propelled through a segment of gut that contains the aganglionic tissue? If CMMCs are abnormal, does transplantation restore function? These experiments would be exciting if they were to work out. The authors have not reported whether they even tried to obtain this critical information.

In summary, this is an excellent and well-conducted study. The limitations of the work are addressed too briefly and only, seemingly in passing, at the end. Limitations should be outlined at the start, and the authors might state then why their study is valuable despite these drawbacks. Peristaltic restoration is the goal of transplantation. The authors need to state whether this goal was addressed but found to be impossible achieve, or whether the authors thought it would be impossible to achieve and thus did not make an attempt to address it. Certainly, the demonstration that peristalsis could actually be improved by transplantation would turn this excellent paper into an outstanding and notable achievement.

Reviewer #2 (Remarks to the Author):

In this paper, Pan and colleagues describe how they succeed to transplant autologous ENS cells into the large intestine of a mouse in which the endogenous neurons were ablated using a genetic diphtheria toxin approach. In a first step of the surgery they remove part of the small intestine from which they derive enteric stem cells, during the same surgery they also inject toxin into the wall of the large intestine, killing the enteric neurons and glia cells as these express DTR. Using the small intestinal cells neurospheres are generated in vitro, which they transduce using lentiviral technology to express either GFP or ChR2. After reintroducing these neurospheres in the aganglionated colon, the authors are able to demonstrate integration in the gut wall and repaired contractility using muscle strip organ bath measurements. The authors cleverly build on previous technology and experience to start bridging the gap between mere lab demonstration of SC integration and clinical application of regenerative approaches. The establishment of an autologous repair approach is an important contribution to the field as it avoids immune reactions as is the case when cells from other animals were used.

Some points need clarification.

Why are the mice kept on a liquid diet? I am left with the impression that the mice do not return to normal chow feeding after the first surgery? Is that correct? I can understand that a liquid diet is necessary at first, to make sure that the anastomosis is not challenged too badly? Or is it also related to the fact that colonic neurons are ablated? In case of the latter, this contrasts with another conclusion in the paper indicating that there are no motility problems in the intestine and that this cannot be used as a readout for repair. Please clarify.

Can the authors provide some estimates (or preferably measurements) of how extensive the area of neuron ablation is? In figure 2 there is just an example, but without quantification it is hard to know how consistent the effect of different injections was.

The injection of ~ 60 neurospheres is mentioned, all of them containing a mixture of glial cells, neurons and stem cells. What fractions of these different types of cells are present in the spheres and do the authors know whether it is mainly the SC (or glia) that colonize the aganglionated gut or do the transplanted (mature) neurons also survive?

A brief description of the morphology or outgrowth characteristics of the transplanted cell network should be provided. Do these cells repopulate the previous ganglion structures (or have these completely disappeared?), or do they grow fairly radially from the site of injection?

In figure 3E, the contractile response to ACh is ~70% higher in DT-AG mice than in control. The authors claim that the contractile responses are similar, based on the fact that they are not significantly different. Given the n=4, I don't think this is sufficiently powered to come to a statistically sound conclusion. A difference of 70% seems substantial and the absence of significance (given low numbers) is not proof of being similar. Do the authors have an idea as to why there is such a difference in responses in the DT-AG mice.

The blue light stimulation should be detailed a bit better. Was the full 200 mW shone onto the tissue? What was the loss of power over the fiberoptic? Is it correct that the light was on for about 30s (as seen in figure 4)? This seems all somewhat excessive? Can the authors comment on this? Furthermore, the response seems to come just after the light stimulation. Would that be an indication that mainly an inhibitory component is activated and that as a result of switching off the light, an off response is elicited?

Line 205: just refer to "full normalization of contractility" rather than "motor activity", the latter would imply that peristaltic measurements have been performed, which is not the case.

It would be easier if the stimulus BLS, ACh and KCl were added as captions to figure 4 B,C, D.

In the abstract, use 'optogenetic stimulation' rather than 'analysis', the optogenetic approach is not analyzed, just used to stimulate. The contractions are analyzed.

Careful proof reading of the MS is warranted as some articles are missing. Eg. line 115 "The" pan-neuronal marker....

Reviewer #3 (Remarks to the Author):

This manuscript describes the first study to demonstrate successful transplantation of autologous enteric neuron stem cells (ENSCs) in a model of focal aganglionosis. The work is an extension of previous studies performed by this group that transplanted enteric neural crest derived cells into mice with focal

aganglionosis induced by diphtheria toxin injection (Bhave et al., 2019). Here, the new and exciting finding is that ENSCs can be cultured from and transplanted back into the same mice, thus reducing potential issues of immunogenicity that can occur when using non-autologous transplants. While the manuscript reports incremental, yet highly significant findings, the paper is missing relevant analyses and methodological details required for accurate interpretation of results and conclusions. These are discussed below, as well as other comments/suggestions.

1. Additional quantitative analyses are needed regarding the engraftment, migration, and differentiation of transplanted ENSCs in vivo:

-To better understand how well transplanted ENSCs repopulated aganglionic regions, the authors should quantify: average area of aganglionosis in DT-AG versus DT-AG + cells mice; average distance that transplanted ENSCs migrated

-To confirm that transplanted ENSCs differentiated into the appropriate neurochemical subtypes of neurons in vivo, the authors should qualitatively show co-expression of GFP and nNOS as was done on those that were cultured on fibronectin-coated glass chambers; an additional marker for excitatory neurons, e.g., ChAT, should also be used, as was done in their previous paper (Bhave et al., 2019). Including quantitative comparison of the ratios of these neurochemical subtypes would confirm that transplanted ENSCs can differentiate into inhibitory and excitatory ENS subtypes at normal proportions.

-Also missing is how many transplanted GFP+ ENSCs differentiated into neurons versus glia.

-Finally, Figure 2 shows findings at 2 weeks after transplantation, whereas the functional assays in Figures 3 and 4 were performed at 3 and 8 weeks after transplantation. Although an immunofluorescence image is shown in Figure 3 that represents the 3 or 8 week time point, the authors should perform the quantitative comparisons mentioned above at these two time points, especially because of the dramatic functional improvements from 3 to 8 weeks shown in Figure 3C.

2. The motility traces shown in Figures 3 and 4 leave me with questions about methodologies used for data collection and analysis that may influence interpretation of results and overall conclusions:

-The example traces in Figure 3A clearly show an increase in the frequency of spontaneous motor complexes caused by the DT injections and this should be quantified and included in the results. There were no differences in spontaneous motility in the group's previous paper; but if there are statistical differences here, the authors should acknowledge the discrepancy in results and offer an explanation.

-In the traces provided in Figure 3A, the evoked contractions occur immediately following stimulation. How much time after stimulation was used to analyze the effect of electrical stimulation? Were repeated trials of stimulation performed (with ample recovery time) to get an average for each animal?

-On a similar note, what was considered baseline and how was it measured? The control trace in Figure 3A does not show motor complexes before stimulation. Did the authors wait for at least one contraction prior to stimulation to use as baseline? Because of the variability in spontaneous contractions between groups (and probably between animals within a group), it is imperative to include precise details of how measurements were made.

3. Experiments in Figure 4 lack a control group: To draw conclusions about whether the transplanted cells are functionally normal, the authors should include a ChR2 cells control group by injecting the same ChR2 virus into the mid-colon of otherwise naïve mice and measuring the contractions evoked by laser stimulation.

4. Individual data points should be added to all graphs.

5. The authors state in their discussion that they allowed necessary time for transplanted ENSCs to establish neuromuscular connections, but that was not shown in their results – Were GFP neuron fibers observed in smooth muscle layers? If so, was it dependent on time? Were there differences between 2, 3 and 8 weeks post transplantation?

6. Line 203 states that improvement in contractile was observed by 2 weeks, but the results mentions that these tests were performed at 3 and 8 weeks.

7. Line 206-207: it would be more accurate to say “...by autologous cell therapy...”

8. There is no mention of the sex of mice used in these studies.

9. Please include the name of the post-hoc test for multiple comparisons when using ANOVA (e.g., Tukey's).

Responses to reviewers' comments

Manuscript ID: NCOMMS-23-01394

Reviewer #1 (Remarks to the Author):

1. This is an excellent paper that reports an important step on the road to devising a clinically useful treatment of enteric neuropathies with transplants of autologous cells. A major strength of the work is that the authors have devised a means of producing a limited degree of colonic aganglionosis that allows mice to survive long enough to permit an autologous graft of enteric neural crest-derived stem cells to be obtained, cultured, marked, transplanted, and enhance neuromuscular function in the region of bowel that was rendered aganglionic. Although the observations are not unexpected, the demonstration is striking in the elegance of the methods and the necessity of going from predictable to demonstrable. A weakness of the study, which the authors acknowledge at the end of the discussion, is that the lesion produced with diphtheria toxin injection into the colon of Wnt1::Cre;R26-iDTR (Wnt1-iDTR) mice, is both limited in size and highly artificial. The aganglionic region, while real, does not actually mimic any of the conditions that prevail, either in mice with a genetic cause of aganglionosis, or in humans with a genetic disorder, such as Hirschsprung disease (HSCR). In the authors' model, enteric neurons express DTR and thus die when exposed to DT. In the natural conditions, neurons and even crest derivatives may not be the only cells that are affected by the abnormality. The microenvironment of the aganglionic colon may itself be abnormal. Abnormalities of the extraganglionic matrix, for example, have been described in humans with HSCR and in murine models of the condition. The aganglionic region in HSCR and its murine models, furthermore, is not denervated. It is heavily innervated by thick nerve trunks that grow into the aganglionic bowel from more proximal ganglionic segments of the gut and from extrinsic ganglia. These nerves too may change the microenvironment, but they also illustrate the need for peristalsis that depends on the presence of a functional ENS. The authors' model, therefore, while aganglionic, is not a precise mimic of the naturally occurring aganglionosis of HSCR. The ability of autologous neural stem cells to survive and differentiate in the colon of Wnt1::Cre;R26-iDTR mice does not in any way suggest that the cells will do the same in a tissue in which endothelin or Ret signaling are abnormal.

We thank the review for such insightful comments. We agree that a more comprehensive discussion of the limitations of the focal aganglionic mouse model would greatly strengthen the paper and we have modified the discussion section accordingly. The revision now includes a summary of the strengths and weaknesses of the model and why we believe that in spite of its limitations the model advances knowledge in the field by allowing us to perform true autologous ENSC studies with tissue harvest and cell transplantation in the same animal, similar to the proposed clinical approach for autologous cell therapy in HSCR patients. We go on to point out that while the focal ablation approach is not a precise mimic of naturally occurring aganglionosis, it is sufficient to permit valid testing of the ability of autologous neural stem cells to form ENS and confer function onto denervated smooth muscle.

2. A second perceived weakness of the authors' approach, again discussed only briefly at the end of the manuscript, is that the autologous stem cells from the small intestine of the Wnt1::Cre;R26-iDTR mice are themselves normal unless they encounter DT. Autologous stem cells from the small intestine of a human with HSCR or a mouse model of that condition, might not be normal and may be affected by the genetic abnormality at the root of the condition. Again, the authors' model does not mimic a natural disease. Because the transplanted cells in the model seem to survive and restore some function, one cannot conclude that similarly transplanted autologous cells will do the same in a patient or even in a mouse model of HSCR.

Regarding this question about testing "normal" autologous stem cells from Wnt1::Cre;R26-iDTR mice, bearing in mind that autologous stem cells obtained from the intestine of HSCR patients or HSCR rodent models might not be normal but in some way affected by the underlying genetic abnormality, we believe there is knowledge to be gained from performing these studies. HSCR is regarded as a neurocristopathy where the primary defect resides in the neural crest cells that give rise to the ENS. What the cellular defects in the enteric neural crest cells actually are, how they result in the failure to colonize the entire length of the gut, and how any coding or non-coding genetic mutations in HSCR patients relates to cellular and molecular defects in these cells, has been intensely studied, but remains complex and not well understood. We believe it is informative to test these "normal" ENSCs by transplant into the focally generated aganglionic region as (i) it is a valid assessment of the ability of autologous neural stem cells to engraft, migrate, extend neurites, become integrated with the neuromuscular circuitry, and confer function onto denervated smooth muscle as mentioned above; (ii) regardless of genetic/cellular defects, in the majority of HSCR patients (those with short segment disease, ~70%+),

most of the gut is colonized by enteric progenitors, with only the very terminal region of the colon remaining aganglionic. This suggests that in HSCR, enteric neural crest cells are “normal” enough to have sufficient migration/proliferation/differentiation capabilities to form the vast majority of the ENS. Consequently, ENSCs harvested from the “normoganglionic” region of HSCR patient bowel likely have sufficient ENS-forming capability to be used for autologous cell therapy. (iii) In support of this previous point we have published data showing that ENSCs obtained from the *Ednrb*^{-/-} mouse model of HSCR have capacity for self-renewal and neuronal differentiation and that the extent of proliferation and differentiation of these *Ednrb*^{-/-} derived cells is not significantly different from that of cells derived from wild type littermates (Hotta et al, *Neurogastroenterology and Motility* 2016).

3. A final problem is that the authors have not tested peristaltic activity or propulsion. They have looked only at neuromuscular function. The authors have shown that destruction of colonic motor neurons removes cholinergic excitation of smooth muscle and that restoration of such neurons can restore cholinergic neural drive to the smooth muscle. These experiments are as elegant as the results are anticipated, but they do not speak to therapeutic efficacy. The authors maintain that they cannot measure peristalsis in the colon after injecting DT into *Wnt1::Cre;R26-iDTR* mice because the lesion is too small; however, they do not say whether or not they tried to do evaluate peristalsis. Many labs have measured colonic migrating motor complexes (CMMCs) ex-vivo in mice, which are essentially peristaltic reflexes driven by the ENS. The authors report that electrically driven contractility of the muscle is absent in aganglionic bowel of their model. Are CMMCs present? Can they be propelled through a segment of gut that contains the aganglionic tissue? If CMMCs are abnormal, does transplantation restore function? These experiments would be exciting if they were to work out. The authors have not reported whether they even tried to obtain this critical information.

The novel focal ENS ablation model of aganglionosis in vivo was first established in our laboratory and published (Bhave et al, *Scientific Reports*, 2019). In this paper, we demonstrated that global ablation of *Wnt1-iDTR* expressing ENS cells by systemic injection of DT resulted in loss of ENS and colonic dysmotility as shown by an absence of CMMCs using spatiotemporal mapping. When we next examined GI transit and colonic motility in mice with focal ablation of ENS using local injection of DT into the colon, we did not observe changes in solid and liquid gut transit times, neither were there changes in CMMC frequency, length, or velocity as determined with spatiotemporal mapping in DT treated and sham operated control mice. Thus, although this focal ablation model is helpful by avoiding animal mortality, CMMCs and colonic peristaltic activity are not affected. We therefore, in the current study, did not assess changes in CMMCs in the focally ablated region post cell transplant as we have previously shown that there are no CMMC defects to rescue.

4. In summary, this is an excellent and well-conducted study. The limitations of the work are addressed too briefly and only, seemingly in passing, at the end. Limitations should be outlined at the start, and the authors might state then why their study is valuable despite these drawbacks. Peristaltic restoration is the goal of transplantation. The authors need to state whether this goal was addressed but found to be impossible achieve, or whether the authors thought it would be impossible to achieve and thus did not make an attempt to address it. Certainly, the demonstration that peristalsis could actually be improved by transplantation would turn this excellent paper into an outstanding and notable achievement.

We appreciate this comment and have added discussion of the study’s limitations to the manuscript. We feel that the current study, with its limitations, contributes to the overall body of work on autologous stem cells, and along with a number of other ongoing studies in our lab using mouse-derived ENSCs transplanted into *Ednrb*^{-/-} mice in vivo, as well as studies with ENSC derived from human patients, helps to advance knowledge in the field and progress toward therapeutic use of autologous ENSCs for HSCR.

Responses to reviewers' comments

Manuscript ID: NCOMMS-23-01394

Reviewer #2 (Remarks to the Author):

In this paper, Pan and colleagues describe how they succeed to transplant autologous ENS cells into the large intestine of a mouse in which the endogenous neurons were ablated using a genetic diphtheria toxin approach. In a first step of the surgery they remove part of the small intestine from which they derive enteric stem cells, during the same surgery they also inject toxin into the wall of the large intestine, killing the enteric neurons and glia cells as these express DTR. Using the small intestinal cells neurospheres are generated in vitro, which they transduce using lentiviral technology to express either GFP or Chr2. After reintroducing these neurospheres in the aganglionated colon, the authors are able to demonstrate integration in the gut wall and repaired contractility using muscle strip organ bath measurements. The authors cleverly build on previous technology and experience to start bridging the gap between mere lab demonstration of SC integration and clinical application of regenerative approaches. The establishment of an autologous repair approach is an important contribution to the field as it avoids immune reactions as is the case when cells from other animals were used.

Some points need clarification.

1. Why are the mice kept on a liquid diet? I am left with the impression that the mice do not return to normal chow feeding after the first surgery? Is that correct? I can understand that a liquid diet is necessary at first, to make sure that the anastomosis is not challenged too badly? Or is it also related to the fact that colonic neurons are ablated? In case of the latter, this contrasts with another conclusion in the paper indicating that there are no motility problems in the intestine and that this cannot be used as a readout for repair. Please clarify.

We apologize for the confusion. As this reviewer pointed out, the mice were given a liquid diet for 7 days postoperatively, which helped to minimize the impact on gut function following anastomosis. However, mice were returned to normal chow from postoperative day 8. The description about the diet and postoperative care has been revised to include these details.

2. Can the authors provide some estimates (or preferably measurements) of how extensive the area of neuron ablation is? In figure 2 there is just an example, but without quantification it is hard to know how consistent the effect of different injections was.

We appreciate the reviewer's comment. We have measured the area of ENS ablation 5 and 10 weeks following DT injection (3 and 8 weeks following cell transplantation) based on immunostaining for anti-Hu antibody, showing that the surface area of the ENS-ablated colonic segments were $44.1 \pm 13.0 \text{ mm}^2$ at 3 weeks ($n=3$) and $55.1 \pm 4.3 \text{ mm}^2$ at 8 weeks ($n=3$). As this reviewer pointed out, the efficacy of ENS ablation clearly relies on the surgical technique for this model. Micro injection of DT to the thin wall of the mouse colon is technically challenging and there is some variability in outcomes as revealed by this data. These quantitative findings have been added to the Results section of the revised manuscript to help clarify the methodology.

3. The injection of ~ 60 neurospheres is mentioned, all of them containing a mixture of glial cells, neurons and stem cells. What fractions of these different types of cells are present in the spheres and do the authors know whether it is mainly the SC (or glia) that colonize the aganglionated gut or do the transplanted (mature) neurons also survive? A brief description of the morphology or outgrowth characteristics of the transplanted cell network should be provided. Do these cells repopulate the previous ganglion structures (or have these completely disappeared?), or do they grow fairly radially from the site of injection?

Thank you very much for these extremely important comments. We believe that neurons that are present in the aganglionic region post-transplant are derived both from progenitors and from neurons present within the neurospheres. We are currently working on describing and quantifying the contributions of the cell populations within neurospheres to the post-transplant ENS and plan to publish this analysis as a separate paper.

Regarding the comments about description of the morphology or outgrowth characteristics of transplanted cell networks, we repeated autologous cell isolation and transplantation using Wnt1-iDTR mice and obtained additional gut samples at 3 and 8 weeks post transplantation. Wholemount IHC was carefully performed and examined thoroughly. We are happy to provide an additional figure panel (new Figure 3) which describes the characteristics of transplanted ENSC-derived neural networks. We have added descriptions of these findings under the subheading, “*Autologous ENSCs form “neoganglia” that contain enteric neuron subtypes with extensive neural fiber projections*” in the Results section of the revised manuscript.

4. In figure 3E, the contractile response to ACh is ~70% higher in DT-AG mice than in control. The authors claim that the contractile responses are similar, based on the fact that they are not significantly different. Given the n=4, I don't think this is sufficiently powered to come to a statistically sound conclusion. A difference of 70% seems substantial and the absence of significance (given low numbers) is not proof of being similar. Do the authors have an idea as to why there is such a difference in responses in the DT-AG mice.

We appreciate the reviewer's comment. We have performed a number of additional experiments to gain statistical power. Updated results are shown in Figure 3E of the revised manuscript in which the contractile responses to ACh in control and DT-induced aganglionic colon are not statistically different.

5. The blue light stimulation should be detailed a bit better. Was the full 200 mW shone onto the tissue? What was the loss of power over the fiberoptic? Is it correct that the light was on for about 30s (as seen in figure 4)? This seems all somewhat excessive? Can the authors comment on this? Furthermore, the response seems to come just after the light stimulation. Would that be an indication that mainly an inhibitory component is activated and that as a result of switching off the light, an off response is elicited?

We understand the concerns about the possible excessive light power to the gut tissue. We used 200 mW as a light source, but only 20 mW/mm² light was applied to the gut tissue, based on measurements using a Power meter. We have revised the description of the light power in the respective sections of the Materials and Methods.

Regarding the timing of responses to the light stimulation, post-stimulation responses are a common phenomenon in such experimental setups using EFS or BLS as a stimulus. We consistently observed quiescence or inhibitory responses while stimulation was being applied and a rebound contraction just after the stimulation was turned off. Underlying mechanisms have also been well-described (McCann et al. Nat Commun 8, 2017; Axelrod et al. Pediatr Res 35, 1994; Traserra et al. Neurogastroenterol Motil 33, 2021) and it has been shown that the inhibitory effect on gut smooth muscle cells mediated by purinergic or nitrergic neurons is conducted faster than the excitatory pathway mediated by cholinergic neurons.

6. Line 205: just refer to “full normalization of contractility” rather than “motor activity”, the latter would imply that peristaltic measurements have been performed, which is not the case.

The sentence in line 205 has been revised accordingly.

7. It would be easier if the stimulus BLS, ACh and KCl were added as captions to figure 4B, C, D.

The stimulus for each preparation has been added to the revised graphs.

8. In the abstract, use ‘optogenetic stimulation’ rather than ‘analysis’, the optogenetic approach is not analyzed, just used to stimulate. The contractions are analyzed.

The sentence in the abstract has been revised accordingly.

9. Careful proof reading of the MS is warranted as some articles are missing. Eg. line 115 “The” pan-neuronal marker...

We have completed careful proofreading of the revised manuscript before re-submission.

Responses to reviewers' comments

Manuscript ID: NCOMMS-23-01394

Reviewer #3 (Remarks to the Author):

This manuscript describes the first study to demonstrate successful transplantation of autologous enteric neuron stem cells (ENSCs) in a model of focal aganglionosis. The work is an extension of previous studies performed by this group that transplanted enteric neural crest derived cells into mice with focal aganglionosis induced by diphtheria toxin injection (Bhave et al., 2019). Here, the new and exciting finding is that ENSCs can be cultured from and transplanted back into the same mice, thus reducing potential issues of immunogenicity that can occur when using non-autologous transplants. While the manuscript reports incremental, yet highly significant findings, the paper is missing relevant analyses and methodological details required for accurate interpretation of results and conclusions. These are discussed below, as well as other comments/suggestions.

1. Additional quantitative analyses are needed regarding the engraftment, migration, and differentiation of transplanted ENSCs in vivo:

1-1. To better understand how well transplanted ENSCs repopulated aganglionic regions, the authors should quantify: average area of aganglionosis in DT-AG versus DT-AG + cells mice; average distance that transplanted ENSCs migrated.

We appreciate this comment from the reviewer. We have measured the area of ENS ablation 5 and 10 weeks following DT injection based on immunostaining for anti-Hu antibody and showed that the surface areas of ENS-ablated colonic segments were $57.1 \pm 8.7 \text{ mm}^2$ at 5 weeks (n=3) and $52.5 \pm 5.0 \text{ mm}^2$ at 10 weeks (n=3) after DT injection. The average cell coverage at 3 and 8 weeks was $6.9 \pm 1.1 \text{ mm}^2$ and $6.1 \pm 1.9 \text{ mm}^2$ respectively (n=3 of each). Finally, we quantified the length of fibers projected from transplanted ENSCs, measured as $4.3 \pm 0.5 \text{ mm}$ and $4.1 \pm 0.5 \text{ mm}$ at 3 and 8 weeks, respectively (n=3 of each). These new results have been added to the Results section of the revised manuscript.

1-2. To confirm that transplanted ENSCs differentiated into the appropriate neurochemical subtypes of neurons in vivo, the authors should qualitatively show co-expression of GFP and nNOS as was done on those that were cultured on fibronectin-coated glass chambers; an additional marker for excitatory neurons, e.g., ChAT, should also be used, as was done in their previous paper (Bhave et al., 2019). Including quantitative comparison of the ratios of these neurochemical subtypes would confirm that transplanted ENSCs can differentiate into inhibitory and excitatory ENS subtypes at normal proportions.

Thank you for these critically important comments. We have repeated further autologous cell transplant experiments and successfully obtained additional results, including quantitative analysis of ENS composition of new neurons derived from transplanted ENSC. The new figure panel (New Figure 3) has been added to the revised manuscript and we have added the description of ENS composition (nNOS/Calretinin enteric neuron subtypes) as below to the Results section in the revised manuscript:

“We also examined the ENS composition of neo ganglia using immunohistochemistry, demonstrating $84.8 \pm 3.2\%$ (n=3) of ENSC-derived neurons within the neo ganglia were nNOS neurons at 3 weeks post-transplant whereas only $17.9 \pm 10.0\%$ (n=3) were calretinin positive (Figure 3F). Statistical comparison to the ENS composition in the wildtype mouse colon demonstrated significant differences in both subtypes ($31.6 \pm 2.8\%$ nNOS neurons in control, n=3 and $47.9 \pm 3.7\%$ calretinin neurons in control, n=3, *** p<0.001, * p<0.05 in Figure 3F). nNOS/calretinin composition appeared to shift closer toward that in controls over time ($70.2 \pm 7.5\%$ nNOS at 8 weeks, n=3 and $40.3 \pm 6.2\%$ calretinin at 8 weeks, n=3, Figure 3F), however, a predominance of nNOS neurons persisted (** p<0.01 in Figure 3F)”.

1-3. Also missing is how many transplanted GFP+ ENSCs differentiated into neurons versus glia.

We performed additional immunostaining of recipient colon where representative images showing differentiation of transplanted ENSCs into S100+ glial cells have been added to the new Figure 3. Regarding the quantitative analysis to determine the neuroglial ratio, however, we encountered technical difficulties in performing these analyses and can not provide this data in the revised manuscript.

1-4. Finally, Figure 2 shows findings at 2 weeks after transplantation, whereas the functional assays in Figures 3 and 4 were performed at 3 and 8 weeks after transplantation. Although an immunofluorescence image is shown in Figure 3 that represents the 3 or 8 week time point, the authors should perform the quantitative comparisons mentioned above at these two time points, especially because of the dramatic functional improvements from 3 to 8 weeks shown in Figure 3C.

We understand this reviewer's concerns. We are happy to provide a new figure (New Figure 3) in which careful and thorough IHC characterization of recipient colon has been included. We have added the description of the new findings, including quantitative comparisons described above (please see comments #1-2 and #1-3), to the Results section of the revised manuscript.

2. The motility traces shown in Figures 3 and 4 leave me with questions about methodologies used for data collection and analysis that may influence interpretation of results and overall conclusions:

Thank you. To help clarify this point we have added the following paragraph in the Materials and Methods part of the manuscript:

"Data acquisition and analysis of organ bath studies"

"Baseline maximum value was obtained from 60 sec of data collected for 1 min prior to EFS or BLS. Maximum changes for contraction were taken from 60 sec of data starting at the time of the stimulus. These were expressed as absolute changes from baseline maximum values. EFS and BLS were repeated 3 times in 5 min intervals, with the maximum response calculated as a mean of 3 responses".

2-1. The example traces in Figure 3A clearly show an increase in the frequency of spontaneous motor complexes caused by the DT injections and this should be quantified and included in the results. There were no differences in spontaneous motility in the group's previous paper; but if there are statistical differences here, the authors should acknowledge the discrepancy in results and offer an explanation.

Thank you for this important comment. As suggested by the reviewer, we have analyzed the baseline activity by measuring the area under the curve (AUC). We added representative traces during the "Spontaneous" period of each group to the revised Figure 3. We also performed quantitative analysis of AUC and results have been added to Figure 3 of the revised manuscript. In the main text of the revised manuscript, we have added the following paragraph in the results section:

"Colonic smooth muscle obtained from control mice showed non-rhythmic, frequent, but low-amplitude contractions (Figure 4A, left panel, "Spontaneous" in Control trace). In contrast, DT-induced aganglionic colon exhibited high-magnitude rhythmic contractions during the baseline recording (Figure 4A, left panel, "Spontaneous" in DT-AG trace). Interestingly, DT-AG colon where cells have been transplanted did not show these high-amplitude rhythmic contractions during the spontaneous phase (Figure 4A, DT-AG + Cells). These baseline activities were quantitatively analyzed by measuring area under the curve (AUC) at random 60-second time periods, demonstrating significantly larger AUC in the aganglionic colon (Figure 4B, 78 ± 21 g.s in DT-AG, $n=4$) compared to ganglionic colon (Figure 4B, 9 ± 3 g.s in Control, $n=4$; $p < 0.001$). These alterations were partially restored by cell transplantation (Figure 4B, 28 ± 6 g.s in DT-AG + Cells, $n=4$; $p < 0.01$)".

Regarding the question about the discrepancy in results from our previous study, we have reported that induction of focal aganglionosis in mouse colon did not alter the propagating colonic motility as examined by spatiotemporal mapping (Bhave et al. 2019). In the current study, we utilized electrophysiological and optogenetic organ bath studies using the transplanted portion of colonic smooth muscle where we could detect changes in contractile responses. Our current, more detailed evaluation successfully demonstrated a significant decline in the contractile events in the aganglionic colonic smooth muscle, which are restored by ENSC transplantation.

2-2. In the traces provided in Figure 3A, the evoked contractions occur immediately following stimulation. How much time after stimulation was used to analyze the effect of electrical stimulation? Were repeated trials of stimulation performed (with ample recovery time) to get an average for each animal?

We have clarified this issue in the Materials and Methods section of the revised manuscript by adding sentences as in our response to Question 2 above.

2-3. On a similar note, what was considered baseline and how was it measured? The control trace in Figure 3A does not show motor complexes before stimulation. Did the authors wait for at least one contraction prior to stimulation to use as baseline? Because of the variability in spontaneous contractions between groups (and probably between animals within a group), it is imperative to include precise details of how measurements were made.

We have now provided additional details in the Materials and Methods section. As suggested by the reviewer, we have analyzed the baseline activity as area under the curve. We added a representative trace in Figure 4A (left panel) and bar graph as Figure 4B.

3. Experiments in Figure 4 lack a control group: To draw conclusions about whether the transplanted cells are functionally normal, the authors should include a ChR2 cells control group by injecting the same ChR2 virus into the mid-colon of otherwise naïve mice and measuring the contractions evoked by laser stimulation.

We performed similar studies while optimizing the optogenetic methodology, including light stimulation only without cells and did not see any change in activity. As we have already confirmed that the laser does not evoke any contractile activity (Rahman et al. Gastroenterol 164, S-94, 2023) in the colonic smooth muscle, we decided not to add these additional study arms.

4. Individual data points should be added to all graphs.

The graphs and descriptions in the main text have been revised accordingly.

5. The authors state in their discussion that they allowed necessary time for transplanted ENSCs to establish neuromuscular connections, but that was not shown in their results - Were GFP neuron fibers observed in smooth muscle layers? If so, was it dependent on time? Were there differences between 2, 3 and 8 weeks post transplantation?

We have repeated cell transplantation studies and successfully generated extensive new data as provided in the new Figure 3. As shown in the new Fig. 3C, transplanted GFP+ cells clearly innervated the smooth muscle layers, although this analysis hasn't been performed over time. Further immunohistochemical analysis in the new Fig. 3F shows that specification of transplanted ENSCs into calretinin expressing excitatory neurons increases with time, which can contribute to the improved contractility of recipient aganglionic colon over time.

6. Line 203 states that improvement in contractile was observed by 2 weeks, but the results mentions that these tests were performed at 3 and 8 weeks.

Apologies for the confusion resulting from this mistake. The description in line 203 has been corrected accordingly.

7. Line 206-207: it would be more accurate to say "...by autologous cell therapy..."

The sentence in line 206-207 has been revised accordingly.

8. There is no mention of the sex of mice used in these studies.

Animals of both genders were used in this study and this has been added to the Materials and Methods.

9. Please include the name of the post-hoc test for multiple comparisons when using ANOVA (e.g., Tukey's).

The sentence describing the statistical analysis performed in this study has been revised accordingly.

REVIEWERS' COMMENTS

Reviewer #1 (Remarks to the Author):

The authors have taken almost all of the suggestions of the reviews to heart and have conscientiously revised their manuscript. No further changes need to be made.

Reviewer #2 (Remarks to the Author):

Thank you for addressing my comments and questions. I have no further remarks. Congratulations with this excellent piece of work.

Reviewer #3 (Remarks to the Author):

I commend and thank the authors for thoughtfully addressing my comments and concerns. The revised manuscript is much improved and provides a more detailed “picture” of the model and effects of autologous transplantation. Some of the new results are quite interesting but raise a few additional concerns and/or warrant more discussion than what has currently been provided.

Question 1: Thank you for performing these critical experiments and providing these important data. The results in new Figure 3 are most interesting, give context to the functional changes shown in Fig 4-5, provide insights into how ENSCs differentiate in this particular environment, and lay the framework for future studies to build upon.

Question 2: Thank you for including quantitative analyses for baseline (or spontaneous) contractile events in the revised manuscript. Further, I understand your explanation that different assays were used in the present study and Bhave et al. However, in light of the significant changes in baseline contractile activity, I have a few additional concerns to address.

First, there are several places in the manuscript that mention “reduced contractile activity in DT-AG group,” but because there was a significant increase in spontaneous contractile activity, the authors need to clearly distinguish spontaneous and EFS-evoked contractions when discussing changes in contractility. The statement “our results demonstrated a significant decline in contractile events in aganglionic smooth muscle” is not accurate without specifying the type of contractile event being referred to. Related to this, the authors should decide whether they want to use the descriptor “baseline” or “spontaneous” and stick with one.

Second, the consistent increase in spontaneous contractile events in DT-AG and partial recovery in DT-AG + cells are important findings that warrant more discussion in the manuscript than what has been provided.

Third, the regular occurrence of these rhythmic contractile events introduces significant refractoriness, i.e., time during which contractions cannot be evoked, and this may explain why EFS could not evoke additional contractions in DT-AG tissue. Because the spontaneous contractions were reduced in DT-AG + cells, EFS was more effective. Thus, the changes in EFS-evoked contractions may be a secondary consequence of increased spontaneous contractions.

Nevertheless, the partial recovery with cell transplantation (DT-AG + cells) suggests that ENSC transplantation was effective at normalizing some aspects of contractility. One explanation is that spontaneous rhythmic contractions emerge in DT-AG because the region lacks inhibitory neurons, leading to disinhibition of myogenic motor patterns that are normally under tonic neuronal inhibition; and in DT-AG + cells, there is enough inhibition (from the newly-transplanted inhibitory neurons) to significantly reduce these spontaneous contractions. The authors can easily address whether the spontaneous contractile events are myogenic in nature by looking to see if they persisted in the presence of TTX. Similar TTX data were presented for EFS-evoked contractions, and because a baseline period was collected, the same analysis should be done for the effect of TTX on spontaneous/baseline contractions.

Questions 3-9: Thank you for addressing my concerns. No additional comments.

Minor:

- Figure 1E: what is the p value?
- Figure 2D: what do the arrows and arrowheads mean?
- Figure 4-5: the colors in the graphs don't match the legend, and the greens are particularly difficult to match up
- Methods – please use the term “sexes” which is more appropriate than “genders” when describing mice.

Responses to reviewers' comments

Manuscript ID: NCOMMS-23-01394

Reviewer #3 (Remarks to the Author):

I commend and thank the authors for thoughtfully addressing my comments and concerns. The revised manuscript is much improved and provides a more detailed "picture" of the model and effects of autologous transplantation. Some of the new results are quite interesting but raise a few additional concerns and/or warrant more discussion than what has currently been provided.

Question 1: Thank you for performing these critical experiments and providing these important data. The results in new Figure 3 are most interesting, give context to the functional changes shown in Fig 4-5, provide insights into how ENSCs differentiate in this particular environment, and lay the framework for future studies to build upon.

Question 2: Thank you for including quantitative analyses for baseline (or spontaneous) contractile events in the revised manuscript. Further, I understand your explanation that different assays were used in the present study and Bhave et al. However, in light of the significant changes in baseline contractile activity, I have a few additional concerns to address.

First, there are several places in the manuscript that mention "reduced contractile activity in DT-AG group," but because there was a significant increase in spontaneous contractile activity, the authors need to clearly distinguish spontaneous and EFS-evoked contractions when discussing changes in contractility. The statement "our results demonstrated a significant decline in contractile events in aganglionic smooth muscle" is not accurate without specifying the type of contractile event being referred to. Related to this, the authors should decide whether they want to use the descriptor "baseline" or "spontaneous" and stick with one.

We appreciate this comment from the reviewer. Sentences have been revised accordingly to discriminate between EFS-induced and spontaneous contractile activity, and for consistency we have used the term "spontaneous" throughout the revised manuscript.

Second, the consistent increase in spontaneous contractile events in DT-AG and partial recovery in DT-AG + cells are important findings that warrant more discussion in the manuscript than what has been provided.

Sustained inhibition of colonic motility provided by tonic inhibitory neuronal output to smooth muscle of colon is required to absorb liquid from luminal contents and store feces in the colon. We observe this phenomenon in the smooth muscle preparations obtained from control mice which is reflected by nonrhythmic, frequent, low-amplitude contractions. Due to the absence of inhibitory neurons in the DT-induced aganglionic colon, we observed high-magnitude rhythmic contractions in tissues from these animals. Interestingly, the aganglionic colon muscles no longer showed obvious high-amplitude rhythmic contractions after cell transplantation, an observation consistent with the idea that the transplanted cells differentiate into neurons, including inhibitory neurons, that help to normalize gut contractile activity. We have added a new paragraph to the revised manuscript (4th paragraph in the Discussion section) discussing the potential mechanisms underlying the consistent increase in spontaneous contractile activity in DT-induced aganglionic colon and its inhibition following cell transplantation.

Third, the regular occurrence of these rhythmic contractile events introduces significant refractoriness, i.e., time during which contractions cannot be evoked, and this may explain why EFS could not evoke additional contractions in DT-AG tissue. Because the spontaneous contractions were reduced in DT-AG + cells, EFS was more effective. Thus, the changes in EFS-evoked contractions may be a secondary consequence of increased spontaneous contractions.

Thank you for raising this issue. Although the amplitude of myogenic contractions is very high in DT-AG colon the muscle still has the capacity to contract further as we see after the addition of Acetylcholine (ACh) (Figure 4E) and KCl to the organ bath. We also observed increased frequency of contractions after addition of ACh, which is independent of the refractory period of each contraction. Therefore, based on these results, we do not believe that EFS-evoked contractions are a secondary consequence of increased spontaneous contractions.

Nevertheless, the partial recovery with cell transplantation (DT-AG + cells) suggests that ENSC transplantation was effective at normalizing some aspects of contractility. One explanation is that spontaneous rhythmic contractions emerge in DT-AG because the region lacks inhibitory neurons, leading to disinhibition of myogenic motor patterns that are normally under tonic neuronal inhibition; and in DT-AG + cells, there is enough inhibition (from the newly-transplanted inhibitory neurons) to significantly reduce these spontaneous contractions. The authors can easily address whether the spontaneous contractile events are myogenic in nature by looking to see if they persisted in the presence of TTX. Similar TTX data were presented for EFS-evoked contractions, and because a baseline period was collected, the same analysis should be done for the effect of TTX on spontaneous/baseline contractions.

Thank you for this comment. as the Reviewer notes, the voltage-gated sodium channel blocker, tetrodotoxin (TTX) is widely used to distinguish between neuronal and myogenic responses in isolated smooth muscles. In the presence of TTX contractions have been observed in the distal colon of many different species including mice and human. The removal of tonic inhibitory neural regulation that normally suppresses this myogenic contractility is thought to be the cause of these contractile responses (Boeckxstaens et al 1993, Middleton et al. 1993}. Similarly, in the presence of TTX we observed contractions in the colon of control and cell transplanted mice, likely due to blocking inhibitory neural input to smooth muscle cells. Consistent with this, there were no changes in the myogenic contractions of DT-AG colon in the presence of TTX. We have provided raw traces of “spontaneous” activity in the presence of TTX, which is shown in the new Fig. 4B. The Results and Discussion sections have been revised accordingly. A new paragraph (4th paragraph in the Discussion Section) includes a description of the myogenic origin of the rhythmic contractile activities observed in this study.

Questions 3-9: Thank you for addressing my concerns. No additional comments.

Minor:

- Figure 1E: what is the p value?

P value was 0.0148. This number has been added to the revised manuscript and figure.

- Figure 2D: what do the arrows and arrowheads mean?

Arrows indicate transplanted cell-derived nNOS-expressing neurons. This statement has been added to the figure legend of the revised manuscript. And arrowheads have been deleted since they are confusing (they indicate transplanted ENSC-derived neurons, but no nNOS expression).

- Figure 4-5: the colors in the graphs don't match the legend, and the greens are particularly difficult to match up.

Thank you, we have revised the colors of bar graphs in the Fig. 4-5.

- Methods – please use the term “sexes” which is more appropriate than “genders” when describing mice.

Thank you, we have revised the manuscript accordingly.